# Epidemiologic Situation of HIV and Monkeypox Coinfection: A Systematic Review

**DOI:** 10.3390/vaccines11020246

**Published:** 2023-01-22

**Authors:** Brando Ortiz-Saavedra, Elizbet S. Montes-Madariaga, Cielo Cabanillas-Ramirez, Niza Alva, Alex Ricardo-Martínez, Darwin A. León-Figueroa, Joshuan J. Barboza, Aroop Mohanty, Bijaya Kumar Padhi, Ranjit Sah

**Affiliations:** 1Universidad Nacional de San Agustín de Arequipa, Arequipa 04000, Peru; 2Unidad de Revisiones Sistemáticas y Meta-Análisis, Tau-Relaped Group, Trujillo 13001, Peru; 3Escuela de Medicina, Universidad Peruana de Ciencias Aplicadas, Lima 15023, Peru; 4Facultad de Medicina Humana, Universidad de San Martín de Porres, Chiclayo 14012, Peru; 5Vicerrectorado de Investigación, Universidad Norbert Wiener, Lima 15046, Peru; 6Department of Microbiology, All India Institute of Medical Sciences, Gorakhpur 273001, India; 7Department of Community Medicine, School of Public Health, Postgraduate Institute of Medical Education and Research, Chandigarh 160012, India; 8Tribhuvan University Teaching Hospital, Institute of Medicine, Kathmandu 44600, Nepal; 9Dr. D. Y. Patil Medical College, Hospital and Research Centre, Dr. D. Y. Patil Vidyapeeth, Pune 411018, India

**Keywords:** monkeypox, HIV, MSM, co-infection, STIs

## Abstract

The most recent monkeypox (Mpox) outbreak is mostly affecting men who have sex with men (MSM) who participate in high-risk sexual behaviors, which is typically the case among human immunodeficiency virus (HIV) carriers, according to clinical and epidemiological statistics. The objective of this research is to determine the epidemiological situation of HIV and smallpox co-infection. Until 1 October 2022, a thorough evaluation of the literature was conducted utilizing the databases PubMed, Embase, Scopus, and Web of Science. Studies were evaluated based on the criteria for selection. Fifty-three studies met the selection criteria. A total of 6345 confirmed cases of monkeypox were recorded, and 40.32% (*n* = 2558) of these cases also had HIV co-infection. In addition, 51.36% (*n* = 3259) of the men (91.44%; *n* = 5802), whose ages ranged from 18 to 71 years, exhibited MSM-specific sexual behaviors. Co-infection with these two viruses can be especially dangerous because it can exacerbate the symptoms of both diseases and make them more difficult to treat. People with HIV are more vulnerable to certain infections, including monkeypox, because their immune systems are weakened. Therefore, it is important that they take measures to prevent infection, such as avoiding contact with infected animals, risky behaviors, and maintaining good hygiene.

## 1. Introduction

The zoonotic disease known as monkeypox (Mpox) is caused by a double-stranded DNA virus belonging to the genus Orthopoxvirus (monkeypox virus) [1,2]. Humans can contract the Mpox virus through direct contact (sexual or skin-to-skin), respiratory droplets, and fomites that have been exposed to the virus [3].

The World Health Organization (WHO) designated the current outbreak of Mpox disease as a Public Health Problem of International Concern on 23 July 2022 [4]. In addition, 83,487 cases were found in 110 countries by 23 December 2022 [5].

The current global outbreak of Mpox continues to affect mainly homosexuals, bisexuals, and men who have sex with men (MSM), with evidence of an increase in the prevalence of human immunodeficiency virus (HIV) and other sexually transmitted infections (STIs) [6]. According to WHO, current epidemiological data show that 51% (13,769/26,992) of confirmed cases of Mpox have HIV [7]; this is because HIV, STIs, and Mpox can be transmitted through sexual contact [7,8].

HIV infection and being immunocompromised may or may not affect the presentation of monkeypox [9]. Although it is reasonable to assume that, because of underlying immunosuppression, the course of monkeypox should be more severe in persons living with HIV, the effects of Mpox in this patient population have yet to be determined [10].

Given the prevalence of HIV in Mpox cases during the current outbreak, the present study aimed to assess the epidemiology of HIV and Mpox co-infection.

## 2. Materials and Methods

### 2.1. Protocol and Registration

This protocol adheres to the standards specified by the Preferred Reporting Items for Systematic Reviews and Meta-Analyses (PRISMA) statement and has been registered in the Prospective International Registry of Systematic Reviews (PROSPERO) database (CRD42022363632) [11].

### 2.2. Eligibility Criteria

To explore the epidemiological situation of HIV and Mpox co-infection, we included primary research articles that had information on patients older than 18 years with serological diagnosis, polymerase chain reaction (PCR), electron microscopy or immunohistochemical findings positive for Mpox and with a current or previous diagnosis of HIV. We included articles published up to 1 October 2022, with study designs of case reports, case series, and observational studies (cross-sectional, cohort, and case-control). Systematic reviews, scoping reviews, narrative reviews, randomized clinical trials, editorials, conference proceedings, letters to the editor that did not present original results, and other studies that did not answer our research question were excluded. No language restriction was established.

### 2.3. Information Sources and Search Strategy

A comprehensive search strategy composed of phrases related to “HIV” and “Monkeypox” was used to search Pubmed, Embase, Scopus, and Web of Science (Table 1). Searches were completed on 1 October 2022, and results were evaluated separately by four different investigators.

### 2.4. Study Selection

Using Rayyan QCRI (https://rayyan.qcri.org/, accessed on 7 October 2022), two writers (B.O.S. and E.S.M.M.) independently reviewed the titles and abstracts in accordance with the inclusion and exclusion criteria. The full texts of chosen pertinent studies were searched for an in-depth examination. Conflicts were settled by consensus and, if necessary, input from a third author (D.A.L.F.). Selected articles were stored using the Endnote 20 program.

### 2.5. Outcomes

Reporting the epidemiological condition of HIV and Mpox co-infection in adult patients was the main result.

### 2.6. Data Collection Process and Data Items

Three researchers independently retrieved data into an Excel spreadsheet from the chosen objects. The following details were taken out: First author, date of publication, study design, country, N° of patients, sex, age, diagnostic test for Mpox, HIV and Mpox co-infection, other sexually transmitted infections (STIs), Acute HIV, antiretroviral treatment (ART), viral load, CD4+ T-cell count, clinical features, location of skin lesions, treatment and outcomes. To guarantee that there were no duplicate articles or material, a fourth investigator reviewed the final list of included articles.

## 3. Results

### 3.1. Study Selection

The search method initially yielded a total of 809 articles. The PRISMA flow chart shows the selection procedure (Figure 1). A total of 437 articles were reviewed after eliminating duplicates (*n* = 372). Fifty-three articles qualified for inclusion in this systematic review after being chosen from 81 after being screened for titles and abstracts [12,13,14,15,16,17,18,19,20,21,22,23,24,25,26,27,28,29,30,31,32,33,34,35,36,37,38,39,40,41,42,43,44,45,46,47,48,49,50,51,52,53,54,55,56,57,58,59,60,61,62,63,64].

### 3.2. Study Characteristics

The studies (*n* = 53) described cases of concurrent Mpox and HIV infection, including the number of cases, HIV infection, history of sexually transmitted diseases, method of Mpox diagnosis, clinical manifestations, location, and progression of skin lesions, CD4+ T-cell count, HIV viral load, treatment, and outcome (Table 2 and Table 3). Table 2 presents a summary of the general features of the publications included in this review [12,13,14,15,16,17,18,19,20,21,22,23,24,25,26,27,28,29,30,31,32,33,34,35,36,37,38,39,40,41,42,43,44,45,46,47,48,49,50,51,52,53,54,55,56,57,58,59,60,61,62,63,64]. A total of 6345 confirmed cases of simian pox were reported, distributed across countries: United States (*n* = 3169) [19,22,28,37,55], Spain (*n* = 937) [18,25,33,42,49,60], Germany (*n* = 869) [15,27,30,31,41,43,54], United Kingdom (*n* = 300) [23,24,29,51,62], France (*n* = 264) [36], Italy (*n* = 90) [13,17,34,35,39,40,56,57,58,59], Nigeria (*n* = 77) [44,45,46], Portugal (*n* = 71) [12,21,52,53], Israel (*n* = 26) [63], Belgium (*n* = 4) [20], Brazil (*n* = 3) [16,38], Romania (*n* = 2) [47,48], Czech Republic (*n* = 2) [14,64], Taiwan (*n* = 1) [32], Greece (*n* = 1) [50], and Australia (*n* = 1) [26] (Table 1). Of the total cases, 40.32% (*n* = 2558) had co-infection between HIV and Mpox [12,13,14,15,16,17,18,19,20,21,22,23,24,25,26,27,28,29,30,31,32,33,34,35,36,37,38,39,40,41,42,43,44,45,46,47,48,49,50,51,52,53,54,55,56,57,58,59,60,61,62,63,64].

### 3.3. Demographical Characteristics and Diagnostic Method for Monkeypox

Males accounted for 91.44% (*n* = 5802) of the total cases registered with Mpox [12,13,14,15,16,17,18,19,20,21,22,23,24,25,26,27,28,29,30,31,32,33,34,35,36,37,38,39,40,41,42,43,44,45,46,47,48,49,50,51,52,53,54,55,56,57,58,59,60,61,62,63,64]. The patients ranged in age from 18 to 71 years. In addition, 51.36% (*n* = 3259) presented sexual behaviors of being MSM [12,13,14,16,17,18,19,20,21,22,23,24,25,26,27,28,29,30,31,32,33,34,35,36,37,38,40,41,42,43,47,48,49,50,51,52,53,54,55,56,57,58,59,60,61,62,63,64]. The most frequent previous or current sexually transmitted infections were 40.32% HIV (*n* = 2558) [12,13,14,15,16,17,18,19,20,21,22,23,24,25,26,27,28,29,30,31,32,33,34,35,36,37,38,39,40,41,42,43,44,45,46,47,48,49,50,51,52,53,54,55,56,57,58,59,60,61,62,63,64], 10.26% Gonorrhea (*n* = 651) [19,24,25,37,49,51,54,56,58,61,63,64], 3.81% Syphilis (*n* = 242) [13,14,15,19,22,24,25,26,27,28,42,45,48,49,51,54,56,58,60,61,64], and less than 1% HSV (*n* = 42) [23,24,25,51,61,64]. Overall, almost all confirmed Mpox cases were diagnosed by PCR [12,13,14,15,16,17,18,19,20,21,22,23,24,25,26,27,28,29,30,31,32,33,34,35,36,37,38,39,40,41,42,43,44,45,46,47,48,49,50,51,52,53,54,55,56,57,58,59,60,61,62,63,64] and only three studies conducted in Nigeria used positive IgM serology [44,45,46] (Table 2).

### 3.4. Clinical Symptoms, Skin Lesion Localization, CD4+ T-Cells, Treatment and Outcomes

The most frequent clinical manifestations in patients with Mpox were: 50% skin lesions (*n* = 3173) [13,14,15,16,17,18,20,21,22,23,24,25,26,27,28,29,32,33,34,35,36,37,38,39,40,41,42,43,44,45,46,47,48,49,50,51,52,53,54,55,56,57,58,59,60,61,63,64], 38.53% fever (*n* = 2445) [12,13,14,16,17,18,21,22,23,24,25,26,27,30,31,32,33,34,35,36,37,39,40,41,42,43,44,45,46,47,48,49,50,51,52,53,54,55,56,60,61,63,64], 35.56% lymphadenopathy (*n* = 2256) [12,13,14,16,17,18,21,23,24,25,26,27,30,31,32,33,35,36,37,38,39,40,41,42,43,44,45,47,48,49,50,51,52,53,54,55,56,57,60,61,63,64], and 23.61% headache (*n* = 1498) [12,16,17,18,22,30,31,33,36,40,41,42,44,45,49,51,53,55,57,60,61](Table 3). The most frequent locations of lesions were: 33.16% genitalia (*n* = 2104) [12,13,15,16,18,21,24,25,26,27,29,30,31,32,33,34,35,36,37,38,40,43,44,47,48,49,50,51,52,53,54,55,57,58,59,60,61,63], 28.51% anus or perianal area (*n* = 1809) [12,13,14,16,17,18,20,21,22,23,24,25,28,29,30,31,32,33,36,38,39,40,42,43,47,48,49,51,53,55,59,60,61,63,64], and 11.49% mouth, lips, or oral mucosa (*n* = 729) [12,15,18,21,24,27,29,30,31,35,38,40,43,44,45,46,48,49,51,55,56,60] (Table 3). The majority of patients received empirical treatment targeting sexually transmitted diseases and symptomatic, and only Mpox-targeted treatment received: cidofovir (*n* = 19) [36,60,61] and tecovirimat (*n* = 18) [15,22,24,27,28,37,61]. In HIV-positive patients, 428 patients were reported to be receiving ART [12,15,16,17,19,20,23,24,25,26,28,29,32,34,37,38,39,40,41,42,43,44,47,48,49,50,51,54,56,57,60,61,62,64], 114 had a CD4+ T-cell count <350/μL [15,19,21,22,27,28,30,31,38,43,44,45], and 15 had a viral load >200 copies/mL [21,24,27,30,31,42,44,48,51,61]. Finally, six deaths were reported, one of these being of an HIV-positive patient with an undetectable viral load and a CD4+ T-cell count of 74/μL who was receiving chemotherapy for diffuse large B-cell lymphoma with metastases to the spine, skull, and liver [38] and two others in HIV-1 positive patients, one who developed sepsis and another with a CD4+ T-cell count < 20/μL and who died after multiple episodes of seizures [44].

## 4. Discussion

Monkeypox emerged this last year as an important epidemiological topic to approach due to the rapid spread of confirmed cases [5]. Likewise, diverse investigations identified the association between Mpox and people living with HIV [65,66]. There have been reports of higher HIV and other STI prevalence in the current worldwide Mpox outbreak, which has largely afflicted gay, bisexual, and MSM people [19]. A theoretical idea that HIV may enhance Monkeypox virus transmission and vice versa was also identified [44]. However, there is still limited scientific information about Mpox co-infection with HIV. For that reason, we summarize the cases of this co-infection in order to have a better epidemiological view. In this study, the epidemiological situation of HIV and Mpox co-infection was determined.

We assessed 53 studies. Our principal findings reveal that most of the population (91.44%) was male. Moreover, the main diagnostic test for Mpox was PCR, and that finding shows the relevance of this test in the diagnosis of Mpox. Another relevant result is that 51.36% of the cases were MSM, which demonstrate the importance to explore this risk factor. The HIV and Mpox co-infection were 40.3%, and the most frequent clinical signs were skin lesions, fever, lymphadenopathy, and headache. Identifying all these characteristics or possible risk factors generates a better prescription of the early system of vaccination for Mpox. The Centers for Disease Control and Prevention (CDC) support the recommendation of the vaccine to people who have already been exposed to Mpox or someone who might be in risk of exposure [67].

The largest number of cases registered with co-infection of Mpox and HIV are male (91.44%), which is consistent with other reviews in which it is reported that the population most affected by Mpox are men [6,68,69,70]. The prevalence of Mpox and HIV co-infection was 40% of the cases. This could be due to the fact that most of the cases occurred in MSM (51%) and that MSM has a greater HIV prevalence than the overall population [19,71]. In addition, HIV-positive patients are more likely to attend a health care facility and have a diagnostic test for Mpox compared to HIV-negative patients [72]. However, the 40% prevalence reported in this systematic review exceeds the prevalence of HIV in MSM in the USA (23%) [19] and Europe (7.7%) [73]. This disparity would suggest that transmissibility could be higher in people with HIV [74]. Mpox can be transmitted by respiratory secretions, skin lesions, contaminated fomites and through seminal fluid [69,75]. Reda et al. [69] found that seminal fluid from Mpox-infected patients had a high Monkeypox virus DNA positivity rate (72.4%), behind the positivity rate of anogenital/rectal lesion samples (74.3%). Therefore, such sexual behaviors in HIV patients could predispose to Mpox infection.

Of the three reported deaths of patients with Mpox and HIV co-infection, two cases had a CD4+ T-cells count < 200/μL. Agrati et al. [76] found rapid activation and expansion of CD4+ and CD8+ T cells with effector memory phenotype and a good Th1 cell response that persisted even after clinical recovery in Mpox patients. However, it was also found that paucisymptomatic patients had a less active T-cell response [76]. This suggests a link between the immune response and clinical severity from Mpox. In addition, it was previously reported that Mpox has the ability to trigger a state of T-cell unresponsiveness via a unique major histocompatibility complex (MHC), the independent mechanism that prevents the activation of CD4+ and TCD8+ T-cell antiviral responses and cytokine production [77,78]. This is probably related to viral dissemination and clinical severity in the infected host. Therefore, a state of immunosuppression, characterized by a low CD4+ T-cell count and response in HIV patients, could be associated with clinical severity, dissemination, and mortality from Mpox infection.

Early and consistent ART delivery according to modern combination regimens reduces viraemia in HIV-infected individuals in a few weeks [79]. The level of viral suppression can be so high that viral evolution is halted and the immune system is restored [79]. In the study by Agrati et al. [76] it was found that the T-cell response in patients with Mpox did not differ according to HIV status. This was due to the fact that patients with Mpox and HIV had a good viroimmunological status. In addition, although there is currently no strong evidence to support the use of antiviral drugs directed against Mpox [80,81], such as tecovirimat or cidofovir, the CDC’s “Interim Guidance for Prevention and Treatment of Monkeypox in Persons with HIV Infection” [82] recommends the use of tecovirimat according to the viroimmunological status of the patient and thus avoid possible complications [83]. Therefore, in patients with HIV and Mpox co-infection it is necessary to start or continue the administration of ART and, if indicated, a therapy directed against Mpox such as tecovirimat. Potential drug interactions between ART and tecovirimat are not grounds for discontinuation of either [82].

### Limitations and Strengths

Among the limitations of this systematic review is that most of the studies correspond to case reports and case series, while longitudinal observational studies are scarce. Therefore, in order to draw reliable conclusions about the severity and mortality of Mpox in HIV patients, it is necessary to have observational studies with an established control group and to control for different confounding factors, such as previous vaccination status or other comorbidities. Likewise, the information reported regarding HIV stage, antiviral therapy regimen, adherence to treatment, CD4+ T-cells counts and viral load is scarce. It would be important to perform subgroup analyses for these variables to determine their influence during disease development in patients with Mpox and HIV. The available research does not allow us to draw conclusions about the severity and mortality of Mpox in HIV patients. In terms of strengths, the current study had a rigorous methodology because it was carried out in accordance with the PRISMA criteria. Similarly, all steps for selecting research were carried out independently by two or more authors.

## 5. Conclusions

HIV and Mpox are spread through sexual contact and are more frequent in those who engage in male-male sexual behavior. Co-infection between HIV and Mpox occurred in 40.32%. Co-infection with these two viruses can be especially dangerous, as it can exacerbate the symptoms of both diseases and make them more difficult to treat.

## Figures and Tables

**Figure 1 vaccines-11-00246-f001:**
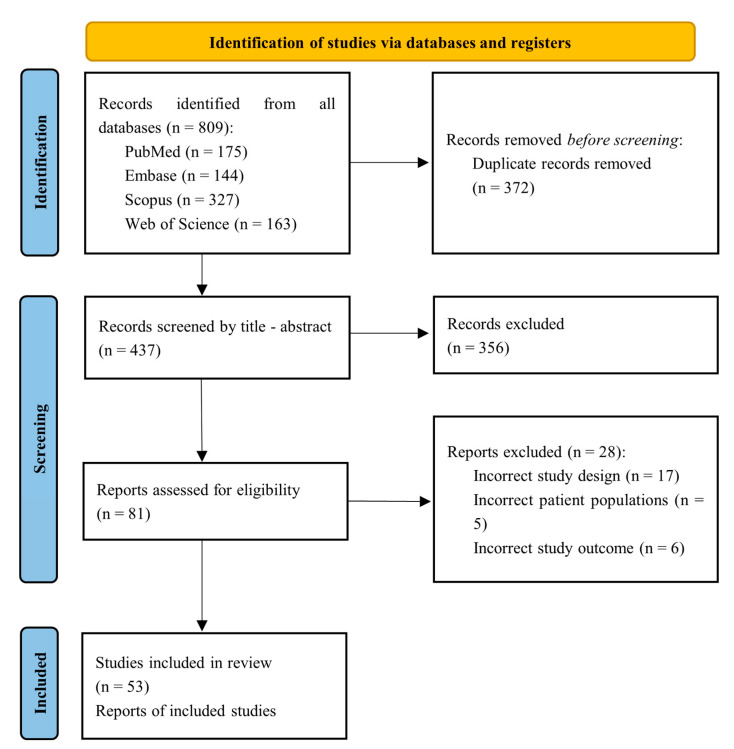
PRISMA flowchart summarizing the process of choosing the studies.

**Table 1 vaccines-11-00246-t001:** Search approach for the literature. * NS: Not Specified.

Base	Search Strategy
PUBMED	#1 “Monkeypox” [MH] OR “Monkeypox” [All Fields] OR “Monkeypox virus” [MH] OR “Monkeypox virus” [All Fields] OR “Monkeypoxvirus*” [TIAB] #2 “HIV” [MH] OR “HIV” [All Fields] OR “Human Immunodeficiency Virus*” [All Fields] OR “Acquired Immunodeficiency Syndrome” [MH] OR “Acquired Immunodeficiency Syndrome” [All Fields] OR “AIDS*” [All Fields] OR “HIV Infection*” [MH] OR “HIV Infection*” [All Fields] OR “HIV Coinfection*” [All Fields] OR “HIV Long-Term Survivors” [MH] OR “HIV Long-Term Survivor*” [All Fields] OR “Sexually Transmitted Diseases” [MH] OR “Sexually Transmitted Disease*” [All Fields] OR “Venereal Disease*” [All Fields] OR “STD” [TIAB] OR “Sexually Transmitted Infection*” [TIAB] OR “STI” [TIAB] OR “Sexual Behavior” [MH] OR “Sexual Behavior” [All fields] OR “MSM” [All fields] OR “Men Who Have Sex With Men” [All fields] #3 = #1 AND #2
SCOPUS	#1 TITLE-ABS-KEY (“Monkeypox” OR “Monkeypox virus” OR “Monkey Pox” OR “Monkeypoxvirus*”) #2 TITLE-ABS-KEY (“HIV*” OR “Human Immunodeficiency Virus*” OR “Acquired Immunodeficiency Syndrome” OR “AIDS*” OR “HIV Infection*” OR “HIV Coinfection*” OR “HIV Long-Term Survivor*” OR “Sexually Transmitted Disease*” OR “Venereal Disease*” OR “STD*” OR “Sexually Transmitted Infection*” OR “STI*” OR “Sexual Behavior” OR “MSM” OR “Men Who Have Sex With Men”) #3 = #1 AND #2
WEB OF SCIENCE	#1 ALL = (“Monkeypox” OR “Monkeypox virus” OR “Monkey Pox” OR “Monkeypoxvirus*”) #2 ALL = (“HIV*” OR “Human Immunodeficiency Virus*” OR “Acquired Immunodeficiency Syndrome” OR “AIDS*” OR “HIV Infection*” OR “HIV Coinfection*” OR “HIV Long-Term Survivor*” OR “Sexually Transmitted Disease*” OR “Venereal Disease*” OR “STD*” OR “Sexually Transmitted Infection*” OR “STI*” OR “Sexual Behavior” OR “MSM” OR “Men Who Have Sex With Men”) #3 = #1 AND #2
EMBASE	#1 ‘monkeypox’/exp OR ‘monkeypox’ #2 ‘human immunodeficiency virus’ #3 = #1 AND #2

**Table 2 vaccines-11-00246-t002:** Characteristics of included studies.

Author	Year	Design	Country	N° of Participants	Age	Sex (M/F)	Diagnostic Method for Mpox	HIV and Mpox Coinfection	Previous STIs	Sexual Behavior
Alpalhão M, et al. [12]	2022	Case series	Portugal	42	HIV: 37.7 ± 9.2 ^a^	M (*n* = 42)	NR	22	NR	MSM (*n* = 37)
					Without-HIV: 32.5 ± 8.1 ^a^					
Antironi A, et al. [13]	2022	Case reports	Italy	4	Median: 30	M (*n* = 4)	RT-PCR	2	Syphilis (*n* = 3), HBV (*n* = 1), HCV (*n* = 1)	MSM (*n* = 4)
Boesecke C, et al. [15]	2022	Case report	Germany	1	40	M (*n* = 1)	RT-PCR	1	Syphilis	NR
Bížová B, et al. [14]	2022	Case report	Czech Republic	1	34	M (*n* = 1)	RT-PCR	1	Syphilis	MSM (*n* = 1)
Brites C, et al. [16]	2022	Case reports	Brazil	2	37 31	M (*n* = 2)	RT-PCR	1	NR	MSM (*n* = 2)
Brundu M, et al. [17]	2022	Case report	Italy	1	35	M (*n* = 1)	RT-PCR	1	NR	MSM (*n* = 1)
Catála A, et al. [18]	2022	Cohort study	Spain	185	38.7 ± 8.2 ^a^	M (*n* = 185)	RT-PCR	78	Yes (*n* = 140) *	MSM (*n* = 184)
Curran KG, et al. [19]	2022	Cohort study	USA	1969	35 (30–42) ^b^	M (*n* = 1466) F (*n* = 10)	RT-PCR	755	Gonorrhea (*n* = 546), Chlamydia (*n* = 489), and Syphilis (*n* = 165)	NR
de Baetselier I, et al. [20]	2022	Cases series	Belgium	4	Range: 30–50	M (*n* = 4)	RT-PCR	3	Yes (*n* = 3) *	MSM (*n* = 3)
de Sousa D, et al. [21]	2022	Case report	Portugal	1	24	M (*n* = 1)	RT-PCR	1	No	MSM (*n* = 1)
Perez-Duque M, et al. [53]	2022	Cases series	Portugal	27	33 (22–51) ^c^	M (*n* = 27)	RT-PCR	14	NR	MSM (*n* = 18)
Gandrakota N, et al. [22]	2022	Case report	USA	1	34	Male-to-Female transgender (*n* = 1)	RT-PCR	1	Neurosyphilis	MSM (*n* = 1)
Gedela K, et al. [23]	2022	Cases series	UK	2	Range: 30–40	M (*n* = 2)	RT-PCR	1	HSV (*n* = 2), and Chlamydia (*n* = 1)	MSM (*n* = 2)
Girometti N, et al. [24]	2022	Cohort study	UK	54	41 (34–45) ^b^	M (*n* = 54)	RT-PCR	13	Syphilis (*n* = 14), HSV (*n* = 24) and Gonorrhea (*n* = 13)	MSM (*n* = 54)
Gomez-Garberi M, et al. [25]	2022	Cases series	Spain	14	42 (20–56) ^c^	M (*n* = 14)	RT-PCR	8	Chlamydia (*n* = 2) Syphilis (*n* = 1), Gonorrhea (*n* = 1), Mycoplasma, HSV-2 (*n* = 1)	MSM (*n* = 10)
Hammerschlag Y, et al. [26]	2022	Case report	Australia	1	30	M (*n* = 1)	RT-PCR	1	Syphilis (*n* = 1)	MSM (*n* = 1)
Heskin J, et al. [29]	2022	Case reports	UK	2	NR	M (*n* = 2)	RT-PCR	1	Negative	MSM (*n* = 2)
Hermanussen L, et al. [27]	2022	Case series	Germany	3	44 (31–54) ^c^	M (*n* = 3)	RT-PCR	1	Syphilis (*n* = 1)	MSM (*n* = 2)
Hernandez LE, et al. [28]	2022	Case report	USA	1	37	M (*n* = 1)	RT-PCR	1	Syphilis (*n* = 1)	NR
Hoffman C, et al. [31]	2022	Cohort study	Germany	546	39 (20–69) ^c^	M (*n* = 546)	RT-PCR	256	Yes (*n* = 286) *	MSM (*n* = 546)
Hoffmann C, et al. [30]	2022	Cohort study	Germany	301	39 (20–64) ^c^	M (*n* = 301)	RT-PCR	141	Yes (*n* = 177) *	MSM (*n* = 301)
Huang S, et al. [32]	2022	Case report	Taiwan	1	24	M (*n* = 1)	RT-PCR	1	NR	MSM (*n* = 1)
Iñigo-Martínez J, et al. [33]	2022	Case series	Spain	508	35 (18–67) ^c^	M (*n* = 503) F (*n* = 5)	RT-PCR	225	NR	MSM (*n* = 397)
Lapa D, et al. [34]	2022	Case report	Italy	1	39	M (*n* = 1)	RT-PCR	1	NR	MSM (*n* = 1)
Loconsole D, et al. [35]	2022	Case series	Italy	10	36 (25–71) ^c^	M (*n* = 8) F (*n* = 2)	RT-PCR	4	NR	MSM (*n* = 6), heterosexual intercourse (*n* = 3)
Mailhe M, et al. [36]	2022	Cohort study	France	264	35 (30–41) ^b^	M (*n* = 262) F (*n* = 1) Trans (*n* = 1)	RT-PCR	73	Yes (*n* = 209) *	MSM (*n* = 245)
Matias WR, et al. [37]	2022	Cases series	USA	3	20 (20–40) ^c^	M (*n* = 3)	RT-PCR	1	Gonococcal urethritis (*n* = 1)	MSM (*n* = 3)
Rodrigues Menezes Y, et al. [38]	2022	Case report	Brazil	1	41	M (*n* = 1)	RT-PCR	1	NR	MSM (*n* = 1)
Mileto D, et al. [39]	2022	Case report	Italy	1	33	M (*n* = 1)	RT-PCR	1	No	NR
Moschese D, et al. [40]	2022	Case series	Italy	32	38 (34–42) ^b^	M (*n* = 32)	RT-PCR	17	NR	MSM (*n* = 32)
Noe S, et al. [41]	2022	Case report	Germany	2	26 32	M (*n* = 2)	RT-PCR	1	NR	MSM (*n* = 2)
Nolasco S, et al. [42]	2022	Case report	Spain	1	36	M (*n* = 1)	RT-PCR	1	Syphilis	MSM (*n* = 1)
Norz D, et al. [43]	2022	Cohort study	Germany	10	Range: 20–50	M (*n* = 16)	RT-PCR	2	NR	MSM (*n* = 16)
Ogoina D, et al. [44]	2020	Cohort study	Nigeria	40	32 (28–54) ^c^	M (*n* = 31) F (*n* = 9)	RT-PCR and IgM serology	9	NR	NR
Ogoina D, et al. [45]	2018	Case series	Nigeria	21	29 (6–45) ^c^	M (*n* = 17) F (*n* = 4)	RT-PCR and IgM serology	2	Syphilis (*n* = 2/8)	NR
Ogoina D, et al. [46]	2022	Case series	Nigeria	16	28 (22–43) ^c^	M (*n* = 12) F (*n* = 6)	RT-PCR and IgM serology	3	Yes (*n* = 4) *	Heterosexual intercourse (*n* = 16)
Oprea C, et al. [47]	2022	Case report	Romania	1	26	M (*n* = 1)	RT-PCR	1	No	MSM (*n* = 1)
Oprea C, et al. [48]	2022	Case report	Romania	1	30	M (*n* = 1)	RT-PCR	1	Syphilis (*n* = 1)	MSM (*n* = 1)
Orviz E, et al. [49]	2022	Descriptive	Spain	48	35 (29–44) ^b^	M (*n* = 48)	RT-PCR	19	Gonorrhea (*n* = 6), Syphilis (*n* = 4)), and *Mycoplasma genitalium* (*n* = 1)	MSM (*n* = 42)
Paparizos V, et al. [50]	2022	Case report	Greece	1	59	M (*n* = 1)	RT-PCR	1	No	MSM (*n* = 1)
Patel A, et al. [51]	2022	Descriptive	UK	197	38 (32–42) ^b^	M (*n* = 197)	RT-PCR	70	Gonorrhea (*n* = 43/161), Chlamydia (*n* = 13/161), Syphilis (*n* = 6/163), and HSV (*n* = 11/157)	MSM (*n* = 197)
Patrocinio-Jesus R, et al. [52]	2022	Case report	Portugal	1	31	M (*n* = 1)	RT-PCR	1	No	MSM(*n* = 1)
Pfäfflin F, et al. [54]	2022	Cases series	Germany	6	Range: 21–50	M (*n* = 6)	RT-PCR	2	Gonorrhea (*n* = 3) and Syphilis (*n* = 1)	MSM (*n* = 6)
Philpott F, et al. [55]	2022	Descriptive	USA	1195	35 (30–41) ^b^	M (*n* = 1178) F (*n* = 5) Transgender man (*n* = 3) Transgender woman (*n* = 5)	RT-PCR	490	NR	MSM (*n* = 337)
Pisano L, et al. [57]	2022	Case report	Italy	1	45	M (*n* = 1)	RT-PCR	1	NR	MSM (*n* = 1)
Pipitò L, et al. [56]	2022	Case reports	Italy	2	45 69	M (*n* = 2)	RT-PCR	2	Syphilis (*n* = 2), Gonorrhea (*n* = 1), and HCV (*n* = 1)	MSM (*n* = 2)
Quattri E, et al. [58]	2022	Case reports	Italy	2	35 29	M (*n* = 2)	RT-PCR	1	Syphilis (*n* = 2) and Gonorrhea (*n* = 1)	MSM (*n* = 2)
Raccagni AR, et al. [59]	2022	Cases series	Italy	36	41.5 (31.25–35.5) ^b^	M (*n* = 36)	RT-PCR	15	Yes (*n* = 4) *	MSM (*n* = 36)
Tarin-Vicente EJ, et al. [60]	2022	Cohort study	Spain	181	37 (31–42) ^b^	M (*n* = 175) F (*n* = 6)	RT-PCR	72	Syphilis (*n* = 13) and Chlamydia (*n* = 10)	MSM (*n* = 166) and MSW (*n* = 15)
Thornhill JP, et al. [61]	2022	Cases series	16 countries	528	38 (18–68) ^c^	M (*n* = 527) Trans (*n* = 1)	RT-PCR	218	Gonorrhea (*n* = 32/377), Chlamydia (*n* = 20/377), Syphilis (*n* = 33/377), HSV (*n* = 3/377), Lympho-granuloma venereum (*n* = 2/377), Chlamydia and Gonorrhea (*n* = 5/377)	Heterosexual (*n* = 9) Homosexual (*n* = 509) Bisexual (*n* = 10)
Vusirikala A, et al. [62]	2022	Descriptive	UK	45	40 (32–43) ^b^	M (*n* = 45)	RT-PCR	11	Yes (*n* = 27) *	MSM (*n* = 45)
Yakubovsky M, et al. [63]	2022	Descriptive	Israel	26	34 (24–53) ^c^	M (*n* = 26)	RT-PCR	7	Gonorrhea (*n* = 3) and *C. trachomatis* (*n* = 3)	MSM (*n* = 26)
Zlámal M, et al. [64]	2022	Case report	Czech Republic	1	38	M (*n* = 1)	RT-PCR	1	HSV, Syphilis, Chlamydia, Gonorrhea, and HCV	MSM (*n* = 1)

Mpox: Monkeypox; UK: United Kingdom; USA: United States of America; MSM:  men who have sex with men; STI: sexually transmitted infection; HIV: human immunodeficiency virus; HBV: hepatitis B virus, HCV: hepatitis C virus, HSV: herpes simplex virus, RT-PCR: Polymerase chain reaction with reverse transcriptase; M/F: Male/Female; NR: No report. ^a^ Media ± SD. ^b^ Median (IQR). ^c^ Median (Range). * NS: Not Specified.

**Table 3 vaccines-11-00246-t003:** Characteristics of eligible studies. HIV status, clinical manifestations, localization, antiretroviral therapy, viral load, CD4+ T-cell count, treatment and outcomes.

Author	N° of Patients	HIV and Mpox Coinfection	Clinical Manifestations	Localization of Skin Lesions	Antiretroviral Therapy	Mpox and Acute HIV	HIV Viral Load	CD4+ T-Cell Count (cells/μL)	Treatment	Outcome
Alpalhão M, et al. [12]	42	22	Fever (*n* = 22), myalgias/arthralgias (*n* = 23), headache (*n* = 21), lymphadenopathy (*n* = 28)	Genital (*n* = 28), perianal (*n* = 22), and perioral (*n* = 12)	Yes (*n* = 22)	No (*n* = 1)	NR	NR	NR	Recovered (*n* = 42)
Antinori A, et al. [13]	1	Yes	Skin lesions and lymphadenopathy	Genital, thorax and calf area	Yes	No (*n* = 1)	NR	NR	Ciprofloxacin, acyclovir, and benzylpenicillin	Recovered
	2	No	Skin lesions, fever, asthenia, and lymphadenopathy	Anal, back, legs and foot sole					Anti-inflammatory and antihistaminic drugs	Recovered
	3	Yes	Skin lesions and fever	Anal, head, thorax, legs, arms, hand, and genital area	Yes	No (*n* = 1)	NR	NR	NR	Recovered
	4	No	Skin lesions, myalgia	Genital and pubic area					NR	Recovered
Boesecke C, et al. [15]	1	Yes	Skin lesions	Nose, penis, and oral mucosa	Bictegravir/emtricitabine/tenofovir alafenamide	Yes (*n* = 1)	NR	127	Oral tecovirimat 600 mg bid for 7 days and ceftriaxone 2 g intravenous for 10 days	Recovered
Bížová B, et al. [14]	1	Yes	High fever, chills, lymphadenopathy, rash, painless perianal erosions, painless ulceration on his left tonsil, and perianal umbilicated papules	Perianal and left side of the body	NR	No (*n* = 1)	NR	NR	Cephalosporins	Recovered
Brites C, et al. [16]	1	Yes	Skin lesions, fever, chills, myalgia, lymphadenopathy, and urethral burning sensation during urination	Forehead, nose, thorax, left leg, glans, and scrotal sac	Lamivudine/tenofovir/efavirenz	No (*n* = 1)	Undetectable	604	Doxycycline and ceftriaxone	Recovered
	2	NR	Skin lesions, fever, headache, back pain, and lymphadenopathy	Legs, trunk, hands, and perianal area	-	-	-	-	-	Recovered
Brundu M, et al. [17]	1	Yes	Skin lesions, fever, lymphadenopathy, chills, myalgia, headache, malaise, sore throat, and episodes of rectal bleeding	Perianal, abdomen, chest, and back	Darunavir/cobicistat/emtricitabine /tenofovir alafenamide	Yes (*n* = 1)	NR	NR	Analgesics	Recovered
Catála A, et al. [18]	185	78	Skin lesions (*n* = 185), lymphadenopathy (*n* = 104), fever (*n* = 100), asthenia (*n* = 81), myalgia (*n* = 81), headache (*n* = 59), proctalgia (*n* = 40), throat ache (*n* = 34), arthralgia (*n* = 21), lumbar pain (*n* = 12), and oral ulcer (*n* = 10)	Genital (*n* = 98), face (*n* = 72), arms (*n* = 70), perianal (*n* = 62), legs (52), thorax (*n* = 47), pubis (*n* = 30), abdomen (*n* = 29), back (*n* = 28), mouth (*n* = 26), plantar (22), palmar (*n* = 12), and eyelids (*n* = 2)	NR	NR	Detectable viral load (*n* = 63)	CD4 count: 698 (549–930) ^a^ CD4 nadir: 396 (249–575) ^a^	NR	Recovered (*n* = 185)
Curran KG, et al. [19]	1969	755	NS	NR	Yes (*n* = 713)	Yes (*n* = 19)	<200 copies/mL (*n* = 618)	639 (452–831) ^a^ <350 (*n* = 91)	NR	Recovered (*n* = 1969)
de Baetselier I, et al. [20]	4	3	Asymptomatic (*n* = 3), Painful vesicular perianal rash (*n* = 1)	Perianal (*n* = 1)	Yes (*n* = 3)	NR	Viral load <20 µL (*n* = 3)	>350 (*n* = 3)	NR	Recovered (*n* = 4)
de Sousa D, et al. [21]	1	Yes	Skin lesions, fatigue, anal pain, lymphadenopathy, and fever	Perianal, genital, mouth, face, and trunk	No (*n* = 1)	Yes (*n* = 1)	>10,000,000 copies/mL	208	Paracetamol, tramadol, and fusidic acid cream	Recovered
Perez-Duque M, et al. [53]	27	14	Exanthema (*n* = 14), inguinal lymphadenopathy (*n* = 14), fever (*n* = 13), genital ulcers (*n* = 6), genital vesicles (*n* = 6), asthenia (*n* = 7), headache (*n* = 7), and myalgia (*n* = 5)	Genital (*n* = 6), anal (*n* = 4)	NR	NR	NR	NR	NR	Recovered (*n* = 27)
Gandrakota N, et al. [22]	1	Yes	Skin lesions, anal pain, headache, fever, photophobia, neck stiffness, and bilateral lower extremity weakness	Perianal	Irregular	No (*n* = 1)	NR	200	Tecovirimat, penicillin, vancomycin, ceftriaxone, ampicillin, doxycycline, and dexamethasone	Recovered
Gedela K, et al. [23]	2	1	Myalgia (*n* = 2), fever (*n* = 2), rectal pain (*n* = 2), lymphadenopathy (*n* = 2), skin lesions (*n* = 1), and throat pain (*n* = 1)	Perianal (*n* = 1)	Yes (*n* = 1)	NR	NR	NR	Aciclovir (*n* = 2), paracetamol (*n* = 2), topical lidocaine (*n* = 2) ibuprofen (*n* = 1), codein (*n* = 1), and morphine (*n* = 1)	Recovered (*n* = 2)
Girometti N, et al. [24]	54	13	Skin lesions (*n* = 54), Fatigue (*n* = 36), fever (*n* = 31), lymphadenopathy (*n* = 30), myalgia (*n* = 16), and sore throat (*n* = 11)	Genital (*n* = 33), perianal (*n* = 24), upper and lower extremities (*n* = 27), facial (*n* = 11), oropharyngeal (*n* = 4), and torso (*n* = 14)	Yes (*n* = 13)	Yes (*n* = 2)	<50 copies/mL (*n* = 11), 200–500 copies/mL (*n* = 2)	>500 (*n* = 13)	Ceftriaxone (*n* = 3), doxycyclin (*n* = 2), metronidazole (*n* = 1), and tecovirimat (*n* = 1)	Recovered (*n* = 54)
Gomez-Garberi M, et al. [25]	14	8	Skin lesions (*n* = 14), lymphadenopathy (*n* = 8), penile oedema (*n* = 6), fever (*n* = 5), malaise (*n* = 4), proctalgia (*n* = 1), and rectal itching (*n* = 1)	Genital (*n* = 12), inguinal (*n* = 1), and perianal (*n* = 1)	Yes (*n* = 8)	Yes (*n* = 1)	NR	Median: 663	Antihistamines, analgesics, and nonsteroidal anti-inflammatory drugs (*n* = 14) and surgical (*n* = 2)	Recovered (*n* = 14)
Hammerschlag Y, et al. [26]	1	Yes	Skin lesions, fever, lymphadenopathy and general malaise	Penis, trunk, face, extremities, hand, calf, and nasal throat	Abacavir/lamivudine/dolutegravir	No (*n* = 1)	< 100 copies/mL	700	Ceftriaxone, doxycycline, cephalexin, and oral analgesia	Recovered
Heskin J, et al. [29]	2	1	Skin lesions (*n* = 2)	Genital (*n* = 1), pubic (*n* = 1), oral and buccal mucous membranes (*n* = 1), perioral (*n* = 1), and perianal (*n* = 1)	Yes (*n* = 1)	No (*n* = 1)	NR	NR	Oral antiviral, antibacterial medication (ceftriaxone) (*n* = 2)	Recovered (*n* = 2)
Hermanussen L, et al. [27]	1	No	Skin lesions, fever, and malaise	Face, hairy scalp, trunk, extremities, oral, palmar, and plantar regions	-	-	-	NR	Tecovirimat, amoxicillin and clavulanic acid	Recovered
	2	No	Skin lesions, lymphadenopathy, and myalgia	Trunk and extremities	-	-	-	NR	Tecovirimat and penicillin	Recovered
	3	Yes	Skin lesions, fever, malaise, and weakness	All over the body, but sparing the genital area	Irregular	No (*n* = 1)	1.29 × 10^6^ copies/mL	50	Tecovirimat	Recovered
Hernandez LE, et al. [28]	1	Yes	Skin lesions	Trunk, upper and lower extremities, groin, and perianal region	Emtricitabine/tenofovir /doravarine/darunavir/cobicistat	No (*n* = 1)	<20 copies/mL	262	Tecovirimat, doxycycline, ceftriaxone, and valacyclovir	Recovered
Hoffman C, et al. [31]	256	256	Fever (*n* = 126), headache and pain in the limbs (*n* = 98), night sweats (*n* = 40), and lymph node swelling (*n* = 95)	Genital (*n* = 110), anal (*n* = 127), oral, perioral, head and neck (*n* = 64), and extremities and/or trunk (*n* = 92)	NR	Yes (*n* = 1)	>50 copies/mL (*n* = 10), >200 copies/mL (*n* = 4)	691 (185–1603) ^b^ <500 (*n* = 42) and <350 (*n* = 7)	NR	Recovered (*n* = 256)
	232 (PrEP User)	0	Fever (*n* = 118), headache and pain in the limbs (*n* = 91), night sweats (*n* = 30), and lymph node swelling (*n* = 95)	Genital (*n* = 114), anal (*n* = 116), oral, perioral, head and neck (*n* = 47), and extremities and/or trunk (*n* = 90)	-	-	-	-	NR	Recovered (*n* = 232)
	58 (Without HIV or PrEP)	0	Fever (*n* = 28), headache and pain in the limbs (*n* = 19), night sweats (*n* = 3), and lymph node swelling (*n* = 23)	Genital (*n* = 43), anal (*n* = 14), oral, perioral, head and neck (*n* = 12), and extremities and/or trunk (*n* = 14)	-	-	-	-	NR	Recovered (*n* = 58)
Hoffmann C, et al. [30]	301	177	Fever (*n* = 168/274), headache and pain in the limbs (*n* = 126/270), night sweats (*n* = 53/266), and lymph node swelling (*n* = 116/264)	Genital (*n* = 146/298), anal (*n* = 152/299), oral, perioral, and head (*n* = 72/296), and extremities and/or trunk (*n* = 122/292)	NR	NR	<50 copies/mL (*n* = 123/130), 50–200 copies/mL (*n* = 4/130), ≥200 copies/mL (*n* = 3/130)	<350 (*n* = 4/127), 350–500 (*n* = 21/127), and ≥500 (*n* = 102/127)	NR	Recovered (*n* = 301)
Huang S, et al. [32]	1	Yes	Skin lesions, lymphadenopathy, fever, sore throat, and myalgia	Face, limbs, trunk, genital, and perianal	Yes (*n* = 1)	No (*n* = 1)	NR	517	NR	Recovered
Iñigo-Martínez J, et al. [33]	508	225	Skin lesions (*n* = 498), fever (*n* = 324), lymphadenopathy (*n* = 311), asthenia (*n* = 238), myalgia (*n* = 185), headache (*n* = 162), odynophagia (*n* = 143), and proctitis (*n* = 81)	Anogenital and/or perineal area (*n* = 359), legs and/or arms (*n* = 222), face (*n* = 177), chest and/or abdomen (*n* = 159), back (*n* = 132), palms and/or plants (*n* = 124)	NR	NR	NR	NR	NR	Recovered (*n* = 508)
Lapa D, et al. [34]	1	Positive	Skin lesions, fever	Head, thorax, legs, arms, hand, and penis	Dolutegravir/lamivudine	No (*n* = 1)	NR	NR	NR	Recovered
Loconsole D, et al. [35]	10	6	Skin lesions (*n* = 10), fever (*n* = 10), shivering and sweating (*n* = 10), and lymphadenopathy (*n* = 10)	Genital (*n* = 7), face (*n* = 6), palms (*n* = 3), arms (*n* = 4), trunk (*n* = 4), back (*n* = 3), and oral (*n* = 2)	NR	NR	NR	NR	NR	Recovered (*n* = 10)
Mailhe M, et al. [36]	264	73	Skin lesions (*n* = 264), lymphadenopathy (*n* = 174), fever (*n* = 171), pharyngitis (*n* = 51), angina (*n* = 41), respiratory signs (*n* = 31), and headaches (*n* = 89)	Genital area (*n* = 135), limbs (*n* = 121), trunk (*n* = 105), perianal (*n* = 100), face (*n* = 88), and palmoplantar area (*n* = 36)	NR	NR	NR	>500 (*n* = 4)	Cidofovir (*n* = 1), valaciclovir (*n* = 1), tobramycin (*n* = 1), ocular dexamethasone (*n* = 1), ganciclovir (*n* = 1), opioids (*n* = 6), acetaminophen (*n* = 9), surgical (*n* = 4)	Recovered (*n* = 264)
Matias WR, et al. [37]	1	No	Skin lesions, lymphadenopathy, fever, chills, and general malaise	Penis, pubis, and arm	-	-	-	-	Tecovirimat	Recovered
	2	Yes	Skin lesions, lymphadenopathy, fever, chills, myalgias, left tonsillar pain, and odynophagia	Forearms and hands	Yes (*n* = 1)	No (*n* = 1)	Suppressed viral load	>500	Tecovirimat	Recovered
	3	No	Skin lesions, lymphadenopathy, malaise, and subjective fevers	Penis, chest, and arm	-	-	-	-	Tecovirimat	Recovered
Rodrigues Menezes Y, et al. [38]	1	Yes	Skin lesions, lymphadenopathy, dyspnea, penis and glans edema, throat pain, diarrhea, weakness, and malaise	Chest, abdomen, back, upper and lower limbs, palms of the hands, soles of the feet, genitalia, perineum, anorectal region, tongue, and oropharynx	Yes	No (*n* = 1)	Undetectable	53	Meropenem and vancomycin	Death
Mileto D, et al. [39]	1	Yes	Skin lesions, fever, lymphadenopathy, asthenia, malaise, faryngodynia, sneezing, and anorexia	Face, both elbows, the trunk, the buttock and the right foot	Dolutegravir/rilpivirine	No (*n* = 1)	<20 copies/mL	771	NR	Recovered
Moschese D, et al. [40]	32	0	Skin lesions (*n* = 7), fever (*n* = 3), lymphadenopathy (*n* = 2), fatigue, asthenia, and malaise (*n* = 9), back pain (*n* = 2), myalgia (*n* = 1), abdominal symptoms (*n* = 2), sore throat (*n* = 2), and headache (*n* = 5)	Genital (*n* = 2), face (*n* = 6), oral (*n* = 2), anal/perianal (*n* = 8), palms (*n* = 3), and soles (*n* = 1)	-	-	-	NR	NR	Recovered (*n* = 15)
		15	Skin lesions (*n* = 9), fever (*n* = 6), lymphadenopathy (*n* = 1), fatigue, asthenia, and malaise (*n* = 11), back pain (*n* = 3), myalgia (*n* = 1), abdominal symptoms (*n* = 2), sore throat (*n* = 4), and headache (*n* = 2)	Genital (*n* = 9), face (*n* = 10), oral (*n* = 1), anal/perianal (*n* = 10), and palms (*n* = 1)	Yes (*n* = 17)	NR	<50 copies/mL (*n* = 16)	678 (526–933) ^a^	NR	Recovered (*n* = 17)
Noe S, et al. [41]	1	Yes	Skin lesions, malaise, fever, arthralgia, myalgia, back pain, headache, dysphagia, and presence of white spots on his tonsils.	Trunk, extremities, and head	Bictegravir/emtricitabine/tenofovir alafenamide	NR	NR	NR	NR	Recovered
	2	No	Skin lesions, fever, fatigue, cough, inguinal lymphadenopathy, and anal pain	Trunk	-	-	-	-	Topical zinc oxide suspension	Recovered
Nolasco S, et al. [42]	1	Yes	Skin lesions, fever, sore throat, fatigue, headache and lymphadenopathy	Perianal, torso, lower limbs, palms, face and glutes	Dolutegravir/abacavir/lamivudine	Yes (*n* = 1)	234,000 copies/mL	812	Sotrovimab	Recovered
Norz D, et al. [43]	10	2	Skin lesions (*n* = 10), lymphadenopathy (*n* = 5), fever (*n* = 3), malaise (*n* = 3), muscle and joint pains (*n* = 2), penile swelling and pain (*n* = 1), and anal pain (*n* = 1)	Genital (*n* = 8), perianal (*n* = 3), oral (*n* = 2), legs (*n* = 2), anal (*n* = 2), arms (*n* = 2), back (*n* = 1), and face (*n* = 1)	Bictegravir/Emtricitabin/Tenofovir alafenamide (*n* = 1) Dolutegravir/lamivudin (*n* = 1)	NR	22 copies/mL (*n* = 1) and not detectable (*n* = 1)	360 (*n* = 1), and 279 (*n* = 1)	Local therapy (*n* = 10), antibiotic therapy (*n* = 2), analgesic (*n* = 2)	Recovered (*n* = 10)
Ogoina D, et al. [44]	40	9	Skin lesions (*n =* 40), fever (*n =* 36), lymphadenopathy (*n =* 35), body aches (*n =* 25), headache (*n =* 19), sore throat (*n =* 18), pruritus (*n =* 15), and conjunctivitis and photophobia (*n =* 9).	Face (97.5%), trunk (92.5%), arms (87.5%), legs (85%), genitalia (67.5%), scalp (62.5%), palms (55%), soles (50%), mouth (37.5%), and eyes (25%)	Yes (*n* = 5)	Yes (*n* = 4)	4798 copies/mL (*n* = 1)	20 (*n* = 1), 55 (*n* = 1), 300 (*n* = 1), 101 (*n* = 1), 354 (*n* = 1), and 357 (*n* = 1)	Symptomatic and supportive care according to the Nigerian interim guidelines for management of HMPOX	HIV: death (*n* = 2) Without-HIV: death (*n* = 3)
Ogoina D, et al. [45]	21	2	Skin lesions (*n* = 21), fever (*n* = 18), itching (*n* = 14), malaise (*n* = 13), headache (*n* = 12), lymphadenopathy (*n* = 13), sore throat (*n* = 9), myalgia (*n* = 5), conjuctivitis (*n* = 4), and diarrhoea (*n* = 1)	NR	NR	Yes (*n* = 2/8)	NR	354 (*n* = 1), and 280 (*n* = 1)	NR	Recovered (*n* = 20) Death (*n* = 1, suicide)
Ogoina D, et al. [46]	16	3	Skin lesions (*n* = 7), and fever (*n* = 9)	NR	NR	NR	NR	NR	NR	NR
Oprea C, et al. [47]	1	Yes	Skin lesions, fever, lymphadenopathy, chills, rectal pain, and dysphagia	Genital, perineal, anal, neck, trunk, and upper and lower limbs	3TC/ABC/DTG	No (*n* = 1)	40 copies/mL	988	Symptomatic treatment, fluids and topic treatment	Recovered
Oprea C, et al. [48]	1	Yes	Skin lesions, fever, lymphadenopathy, malaise, nausea, loss of appetite, and jaundice	Genital, anal, trunk, lumb., face, ear flap, limbs, palms, soles, and oral mucosa	Tenofovir/lamivudine/dolutegravir (adherence problems)	No (*n* = 1)	2820 copies/mL	936	Glucose, arginine, benzathine benzylpenicillin, dexamethasone 8 mg/day, vitamin K and fresh frozen plasma	Recovered
Orviz E, et al. [49]	48	19	Skin lesions (*n* = 45), fever (*n* = 25), asthenia (*n* = 32), myalgia (*n* = 25), lymphadenopathy (*n* = 39), headache (*n* = 25), proctitis (*n* = 13), urethritis (*n* = 7), rash (*n* = 4), nasal congestion (*n* = 4), and cough (*n* = 8)	Genital (*n* = 26), upper extremities (*n* = 20), perianal (*n* = 17), trunk (*n* = 16), facial (*n* = 12), periorally (*n* = 9), lower extremities (*n* = 10), and palms and soles (*n* = 2)	Yes (*n* = 18)	Yes (*n* = 1)	NR	NR	NR	Recovered (*n* = 48)
Paparizos V, et al. [50]	1	Yes	Skin lesions, fever, lymphadenopathy, myalgia and fatigue	Genital	Yes (*n* = 1)	No (*n* = 1)	< 20 copies/mL	648	Topical octenidine and antibiotic ointment	Recovered (*n* = 1)
Patel A, et al. [51]	197	70	Mucocutaneous manifestations (*n* = 197), fever (*n* = 122), lymphadenopathy (*n* = 114), headache (*n* = 49), fatigue/lethargy (*n* = 46), myalgia (*n* = 62), arthralgia (*n* = 21), back pain (*n* = 21), and rectal pain or pain on defecation (*n* = 71)	Face (*n* = 71), trunk (*n* = 70), arms/legs (*n* = 74), hands/feet (*n* = 56), genitals (*n* = 111), anus or perianal area (*n* = 82), and oropharyngeal (*n* = 27)	Yes (*n* = 64/70)	Yes (*n* = 1)	<200 copies/mL (*n* = 55/70)	664 (522–894) ^b^	Paracetamol, ibuprofen, opioids, and lidocaine gel	Recovered (*n* = 197)
Patrocinio-Jesus R, et al. [52]	1	Yes	Skin lesions, lymphadenopathy, fever and sore throat	Genital, face and hands	NR	NR	NR	NR	NR	Recovered (*n* = 1)
Pfäfflin F, et al. [54]	1	Yes	Skin lesions, fever, perianal pain, anal abscess, and lymphadenopathy	Limbs	Yes	NR	NR	870	Ibuprofen	Recovered
	2	No	Skin lesions, fever, malaise, anal pain, and anal fissure	Left arm	-	-	-	-	Metamizole, tramadol, lidocaine topical	Recovered
	3	No	Skin lesions, anal pain, rectal ulcer, and proctitis	Limbs	-	-	-	-	Ibuprofen, metamizole, lidocaine topical	Recovered
	4	No	Skin lesions, fatigue, anal pain, and anal ulcer	Arms, trunk, genital	-	-	-	-	Metamizole, lidocaine topical	Recovered
	5	No	Skin lesions, fever, malaise, myalgia, sweats, anal pain, inflammation of rectum and anal canal	Head, neck, trunk, limbs	-	-	-	-	Metamizole, lidocaine topical	Recovered
	6	Yes	Skin lesions, anal ulcer, myalgia, fever, malaise, anal pain, and proctitis	Legs	Yes	NR	NR	> 500	Metamizole, lidocaine topical	Recovered
Philpott F, et al. [55]	1195	490	Skin lesions (*n* = 1004), fever (*n* = 596), chills (*n* = 550), lymphadenopathy (*n* = 545), malaise (*n* = 531), myalgia (*n* = 507), headache (*n* = 469), rectal pain (*n* = 201), pus or blood in stools (*n* = 184), abdominal pain (*n* = 96), rectal bleeding (*n* = 90), tenesmus (*n* = 90), and vomiting or nausea (*n* = 83)	Genital (*n* = 333), arms (*n* = 284), face (*n* = 276), legs (*n* = 265), perianal (*n* = 225), mouth, lips, or oral mucosa (*n* = 179), palms of hands (*n* = 157), trunk (*n* = 156), neck (*n* = 130), head (*n* = 97), and soles of feet (*n* = 77)	NR	NR	NR	NR	NR	NR
Pisano L, et al. [57]	1	Yes	Skin lesions, lymphadenopathy, asthenia, headache, mild myalgia and cold.	Face, neck, genital, limbs and trunk	Elvitegravir/tenofovir/emtricitabine/cobicistat	No	Undetectable	NR	NR	Recovered (*n* = 1)
Pipitò L, et al. [56]	1	Yes	Skin lesions, fever, malaise, sore throat and painful cervical lymphadenopathy	Oral mucosa and trunk	Yes	NR	Undetectable	NR	NR	Recovered
	2	Yes	Skin lesions, sore throat and painful cervical lymphadenopathy	Oral mucosa and nipple	Yes	NR	Undetectable	NR	NR	Recovered
Quattri E, et al. [58]	1	Yes	Skin lesion	Genital	NR	NR	NR	NR	NR	Recovered
	2	No	Skin lesion	Genital	-	-	-	-	NR	Recovered
Raccagni AR, et al. [59]	36	15	Skin lesions (*n* = 36)	Genital (*n* = 13), rectal (*n* = 18), and cutaneous (*n* = 20)	NR	NR	NR	NR	NR	Recovered (*n* = 36)
Tarin-Vicente EJ, et al. [60]	181	72	Skin lesions (*n* = 181), lymphadenopathy (*n* = 153), influenza-like illness (*n* = 147), fever (*n* = 131), headache (*n* = 96), and sore throat (*n* = 66)	Genital (*n* =100), perianal area (*n* = 66), oral ulcer (*n* = 45), perioral (*n* = 51), hands and feet (*n* = 108), trunk and extremities (*n* = 104)	Yes (*n* = 71)	NR	NR	<500 (*n* = 8)	Cidofovir (*n* = 6)	Recovered (*n* = 181)
Thornhill JP, et al. [61]	528	218	Rash or skin lesions (*n* = 500), fever (*n* = 330), lymphadenopathy (*n* = 295), lethargy or exhaustion (*n* = 216), myalgia (*n* = 165), headache (*n* = 145), pharyngitis (*n* = 113), low mood (*n* = 54), and proctitis or anorectal pain (*n* = 75).	Anogenital area (*n* = 383), trunk or limbs (*n* = 292), face (*n* = 134), palms or soles (*n* = 51), and mucosal lesions (*n* = 217).	Tenofovir-based three-drug regimen (*n* = 102/210), abacavir-based three-drug regimen (*n* = 20/210), zidovudine-based three-drug regimen (*n* = 2/210), two-drug regimen (*n* = 48/210)	NR	< 50 copies/mL (*n* = 180/190), < 200 copies/mL (*n* = 185/190)	680 (513–861) ^a^	Cidofovir (*n* = 12), tecovirimat (*n* = 8), and vaccinia immune globulin (*n* = 1)	Recovered (*n* = 528)
Vusirikala A, et al. [62]	45	11	NR	NR	Yes (*n* = 11)	NR	Undetectable (*n* = 10)	NR	NR	Recovered (*n* = 45)
Yakubovsky M, et al. [63]	26	7	Skin lesions (*n* = 26), proctitis (*n* = 26), fever (*n* = 19), and inguinal lymphadenopathy (*n* = 17)	Anorrectal (*n* = 19), genital (*n* = 11), and other (*n* = 18)	NR	NR	NR	NR	NR	Recovered (*n* = 26)
Zlámal M, et al. [64]	1	Yes	Skin lesions, fever, rash, groin lymphadenopathy, and rectal pain	Anal, perianal, and trunk	Yes (*n* = 1)	No	Undetectable (*n* = 1)	>200	Valaciclovir, ceftriaxone, azithromycin, and metronidazole	Recovered (*n* = 1)

Mpox: Monkeypox; NR: No report. ^a^ Median (IQR). ^b^ Median (Range).

## Data Availability

All data included within the manuscript.

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
