# Peer review of "Epidemiologic Situation of HIV and Monkeypox Coinfection: A Systematic Review"

_vaccines, 2023, doi:10.3390/vaccines11020246_

Round 1

Reviewer 1 Report

To date, data on the severity of Mpox in immunocompromised persons, including those infected with HIV, are scarce. Indeed, it is not well known whether HIV infection increases the risk of developing Mpox after exposure. While this is a topic of interest, there are important limitations.

- The introduction section is confusing.

- Characteristics of included studies. While isolated cases or those involving a very small group of people may provide valuable information, they generally tend to have lower methodological quality and a higher risk of bias. Therefore, it would not be considered appropriate to include them in this study as they do not provide generalisable information on the topic in question. Please, consider removing them (nearly 30 references include <5 patients).

- Other sections are affected by the inclusion of isolated cases or series with a small number of patients.

Author Response

Dear Editor and Reviewers,

Thank you for reviewing our manuscript and giving your positive feedback and suggestions which has improved the manuscript.

We have addressed all the concern raised by editors and reviewer which can be seen in the highlighted manuscript. Also, we have addressed each comment by replying with answer below the comments

REVIEWER #1

To date, data on the severity of Mpox in immunocompromised persons, including those infected with HIV, are scarce. Indeed, it is not well known whether HIV infection increases the risk of developing Mpox after exposure. While this is a topic of interest, there are important limitations.

  1. Reviewer comment: The introduction section is confusing.

Our response: “Thank you very much for your comments. The introduction has been modified and the information has been updated according to recently published evidence.”

  1. Reviewer comment: Characteristics of included studies. While isolated cases or those involving a very small group of people may provide valuable information, they generally tend to have lower methodological quality and a higher risk of bias. Therefore, it would not be considered appropriate to include them in this study as they do not provide generalisable information on the topic in question. Please, consider removing them (nearly 30 references include <5 patients).

Our response: “Thank you for your recommendations. Our systematic review sets out the inclusion criteria for case reports, case series, and observational studies reporting cases of monkeypox-HIV coinfection. Therefore, the requested reports could not be eliminated. Importantly, this systematic review does not present a meta-analysis because of the included studies.”

  1. Reviewer comment: Other sections are affected by the inclusion of isolated cases or series with a small number of patients.

Our response: “Thank you for your comments. Please be informed that the inclusion of case reports and case series corresponds to the inclusion criteria of this systematic review.”

REVIEWER #2

The present article is a well-written, systematic review about HIV and Mpox co-infection. Methods are sound and results are well reported. I would recommend the article for publication in Vaccines.

  1. Reviewer comment: It would be interesting to have meta-regression analysis of possible risk factors for adverse outcome (death).

Our response: “Thank you very much for your recommendations. A meta-regression analysis could not be performed for the present study, due to the inclusion criteria of the present systematic review, which could not meta-analyze the data found.”

  1. Reviewer comment: Tables are a bit long and hard to read. It may be maybe easier for the reader to have a table summarizing main results, reporting single studies results as supplemental material.

Our response: “Thank you for your recommendations. The proposed tables are the result of the 53 articles selected based on the inclusion criteria.” Summarizing all the tables and adding them in supplementary material, could not be done because all the information according to the objective of the present systematic review is described in the results and this could result in the presentation of information that is already in the entire proposed article.

Thank you again for positive comments by which we have improve the manuscript.

If there are more concern, we are ready to make the changes. Hoping for your positive response

 Yours Sincerely,

Dr. Ranjit Sah

Reviewer 2 Report

The present article is a well-written, systematic review about HIV and Mpox co-infection. Methods are sound and results are well reported. I would recommend the article for publication in Vaccines. 

Few suggestions:

-  It would be interesting to have meta-regression analysis of possible risk factors for adverse outcome (death).

- Tables are a bit long and hard to read. It may be maybe easier for the reader to have a table summarizing main results, reporting  single studies results as supplemental material.

Author Response

Dear Editor and Reviewers,

Thank you for reviewing our manuscript and giving your positive feedback and suggestions which has improved the manuscript.

We have addressed all the concern raised by editors and reviewer which can be seen in the highlighted manuscript. Also, we have addressed each comment by replying with answer below the comments

REVIEWER #2

The present article is a well-written, systematic review about HIV and Mpox co-infection. Methods are sound and results are well reported. I would recommend the article for publication in Vaccines.

  1. Reviewer comment: It would be interesting to have meta-regression analysis of possible risk factors for adverse outcome (death).

Our response: “Thank you very much for your recommendations. A meta-regression analysis could not be performed for the present study, due to the inclusion criteria of the present systematic review, which could not meta-analyze the data found.”

  1. Reviewer comment: Tables are a bit long and hard to read. It may be maybe easier for the reader to have a table summarizing main results, reporting single studies results as supplemental material.

Our response: “Thank you for your recommendations. The proposed tables are the result of the 53 articles selected based on the inclusion criteria.” Summarizing all the tables and adding them in supplementary material, could not be done because all the information according to the objective of the present systematic review is described in the results and this could result in the presentation of information that is already in the entire proposed article.

Thank you again for positive comments by which we have improve the manuscript.

If there are more concern, we are ready to make the changes. Hoping for your positive response

 Yours Sincerely,

Dr. Ranjit Sah

Reviewer 3 Report

This article do not report scientifically sound experiments (laboratory and clinical vaccine research) and do not provide any amount of new information in the field of vaccines.

In INTRODUCTION some references are copy from other publications. The data of reference 7 (de Sousa D et al. Human monkeypox coinfection with acute HIV: an exuberant presentation) was published by Perez Duque M et al. Ongoing monkeypox virus outbreak, Portugal, 29 April to 23 May 2022. Eurosurveillance 2022;27(22):1-6. The mathematical model investigating the HIV and monkeypox co-infection was published by Bhunu CP et al. (Modelling HIV/AIDS and monkeypox co-infection. Appl Math Comput 2012;218:9504-518), and was stated in the reference 8 (Ogoina D et al. Clinical course and outcome of human monkeypox in Nigeria). In the same chapter it’s state “…several men with a diagnosis of the human immunodeficiency virus (HIV) developed Mpox…”. This is not the same that we can read in the reference 4 (Ortiz-Martinez Y et al. Monkeypox and HIV/AIDS: When the outbreak faces the epidemic) “Several men who developed monkeypox are living with HIV…”

In MATERIAL and METHODS is written “We included articles published up to October 1, 2022…”, further up “This systematic review comprised 53 articles published up to September 2022, and further still “Table 1 summarizes the basic features of the 53 studies that were considered. All of them were released between 2018 and 2022”. This is confusing.

In DISCUSSION “Table 1 summarizes…” is repetition than described in the chapter of results. Table 2 must be included in SUPPLEMENTARY MATERIALS.

Author Response

LETTER TO REVIEWERS

Dear reviewer, thank you for your comments and recommendations, which have improved the quality of the article. We have addressed all the concern raised by editors and reviewer which can be seen in the highlighted manuscript. Also, we have addressed each comment by replying with answer below the comments

THIRD REVIEWER

  1. Reviewer comment:  This article do not report scientifically sound experiments (laboratory and clinical vaccine research) and do not provide any amount of new information in the field of vaccines.

Our response:  Dear reviewer, thank you for your comments. You are correct in what you mention, the present study does not report scientifically sound experiments (clinical and laboratory vaccine research) and does not provide any amount of new information in the field of vaccines.

The present study addressed the epidemiology of monkeypox and was uploaded in the special section "Monkeypox virus infection: analysis and detection".

That is why this article aimed to evaluate the epidemiology of HIV and Mpox coinfection, which provides valuable information and corresponds to the special section proposed by the journal. The article follows a solid and rigorous methodology, which provides scientific evidence according to the proposed objective.

  1. Reviewer comment:  In INTRODUCTION some references are copy from other publications. The data of reference 7 (de Sousa D et al. Human monkeypox coinfection with acute HIV: an exuberant presentation) was published by Perez Duque M et al. Ongoing monkeypox virus outbreak, Portugal, 29 April to 23 May 2022. Eurosurveillance 2022;27(22):1-6. The mathematical model investigating the HIV and monkeypox co-infection was published by Bhunu CP et al. (Modelling HIV/AIDS and monkeypox co-infection. Appl Math Comput 2012;218:9504-518), and was stated in the reference 8 (Ogoina D et al. Clinical course and outcome of human monkeypox in Nigeria). In the same chapter it’s state “…several men with a diagnosis of the human immunodeficiency virus (HIV) developed Mpox…”. This is not the same that we can read in the reference 4 (Ortiz-Martinez Y et al. Monkeypox and HIV/AIDS: When the outbreak faces the epidemic) “Several men who developed monkeypox are living with HIV…”

Our response:  Thank you very much for your comments. The introduction section was modified and in agreement with the other reviewers who requested a greater focus according to the stated objective. That is why recently published evidence has been added to support our study.

  1. Reviewer comment:  In MATERIAL and METHODS is written “We included articles published up to October 1, 2022…”, further up “This systematic review comprised 53 articles published up to September 2022, and further still “Table 1 summarizes the basic features of the 53 studies that were considered. All of them were released between 2018 and 2022”. This is confusing.

Our response:  Corrected and improved the clarity of the wording. Thank you very much for your comments.

  1. Reviewer comment: In DISCUSSION “Table 1 summarizes…” is repetition than described in the chapter of results. Table 2 must be included in SUPPLEMENTARY MATERIALS.

Our response: The observations indicated in the discussion section were corrected. Tables 2 and 3 could not be included in supplementary material because they are the main data of the article and meet our objective.

Thank you again for positive comments by which we have improve the manuscript.

If there are more concern, we are ready to make the changes. Hoping for your positive response

Yours Sincerely,

Dr. Ranjit Sah

Round 2

Reviewer 1 Report

As it was mentioned, the characteristics of the selected studies (some of them with only 1 case) may condition the conclusions. The authors consider that it should not be taken into account based on their criterion.

Because a systematic review is usually performed to do a meta-analysis, in this article it would be highly recoomended to do it.

Author Response

LETTER TO REVIEWERS

Dear reviewer, thank you for your comments and recommendations, which have improved the quality of the article. In the following paragraphs, we will report on the lifting of comments and acceptance of suggestions.

FIRST REVIEWER

  1. Reviewer comment:  As it was mentioned, the characteristics of the selected studies (some of them with only 1 case) may condition the conclusions. The authors consider that it should not be taken into account based on their criterion.

Because a systematic review is usually performed to do a meta-analysis, in this article it would be highly recommended to do it.

Our response:  Thank you for your comment. We originally intended to meta-analyze, however, the studies do not show the desired heterogeneity to weigh the effects. On the other hand, we decided not to meta-analyze because we are not comparing trigger-factor, but our question is more descriptive, and therefore only a systematic review was proposed.

A meta-analysis is an advance over the systematic review and involves the use of mathematical and statistical approaches to summarize the results of studies used for a systematic review. Systematic review itself is one of the good forms of research for obtaining result in medical science.

Our study presents several reasons for not performing a meta-analysis.

  • The results of the included studies (reports and case series) could have specific biases or errors.
  • The methodology should be changed, and the quality of evidence should be evaluated according to the GRADE system.

             Therefore, I leave it up to the editor's decision. Thank you very much, dear editor.

Reviewer 3 Report

The INTRODUCTION was deeply transformed, sustained by recently published evidence. Section MATERIAL and METHODS was reviewed and corrected. Section DISCUSSION was corrected, avoiding duplications. With the changes done the manuscript was improved.

I suggest a minor correction at the 3rd line, DISCUSSION section “…people living with HIV [65,66]”. Instead “…people living with HIV/AIDS [65,66]”. Ref. UNAIDS Terminology Guidelines 2015.

Author Response

LETTER TO REVIEWERS

Dear reviewer, thank you for your comments and recommendations, which have improved the quality of the article. In the following paragraphs, we will report on the lifting of comments and acceptance of suggestions.

THIRD REVIEWER

  1. Reviewer comment:  The INTRODUCTION was deeply transformed, sustained by recently published evidence. Section MATERIAL and METHODS was reviewed and corrected. Section DISCUSSION was corrected, avoiding duplications. With the changes done the manuscript was improved.

Our response:  Thank you very much, dear reviewer. This has been possible thanks to your recommendations and your review.

  1. Reviewer comment:  I suggest a minor correction at the 3rd line, DISCUSSION section “…people living with HIV [65,66]”. Instead “…people living with HIV/AIDS [65,66]”. Ref. UNAIDS Terminology Guidelines 2015.

Our response:  The terminology was corrected as indicated.

Sir, we have addressed all the concern raised the by the editors and reviewers. If more concern is remaining, we are ready to make changes as per your advice and suggestions.

Thank you again for all your input, guidance and help which has improved the manuscript.

Yours Sincerely,

Dr. Ranjit Sah

MBBS, MD, Infectious Diseases Fellowship, Clinical Research (Harvard Medical School), Global Clinical Scholars Research Training (Harvard Medical School), Systematic Review and Meta-analysis (London School of Hygiene and Tropical Medicine)

Editorial Board Member: Travel Medicine and Infectious Diseases (IF 20), ID Cases and BMC infectious Diseases

Round 3

Reviewer 1 Report

None